# Comparative Transcriptomics of *Chilodonella hexasticha* and *C. uncinata* Provide New Insights into Adaptations to a Parasitic Lifestyle and Mdivi-1 as a Potential Agent for Chilodonellosis Control

**DOI:** 10.3390/ijms241713058

**Published:** 2023-08-22

**Authors:** Xialian Bu, Weishan Zhao, Wenxiang Li, Hong Zou, Ming Li, Guitang Wang

**Affiliations:** 1State Key Laboratory of Freshwater Ecology and Biotechnology, and Key Labatory of Aquaculture Disease Control, Ministry of Agriculture, Institute of Hydrobiology, Chinese Academy of Sciences, Wuhan 430072, China; xialianbu30@163.com (X.B.); zws@ihb.ac.cn (W.Z.); liwx@ihb.ac.cn (W.L.); zouhong@ihb.ac.cn (H.Z.); gtwang@ihb.ac.cn (G.W.); 2University of Chinese Academy of Sciences, Beijing 100049, China; 3Protist 10,000 Genomics Project (P10K) Consortium, Institute of Hydrobiology, Chinese Academy of Sciences, Wuhan 430072, China

**Keywords:** Chilodonellid, energy metabolism, mitochondria, parasitism, mdivi-1

## Abstract

*Chilodonella hexasticha* is a harmful parasitic ciliate that can cause severe damage to fish and high mortalities worldwide. Its congeneric species, *C. uncinata*, is a facultative parasite that not only can be free-living but also can parasitize on fish gills and fins. In this study, single-cell transcriptomes of these two species were assembled and characterized. Numerous enzymes related to energy metabolism and parasitic adaption were identified through annotation in the Non-Redundant (NR), Clusters of Orthologous Genes (COG), Gene Ontology (GO) and Kyoto Encyclopedia of Genes and Genomes (KEGG) databases. The expression of isocitrate dehydrogenase (IDH), cytochrome c oxidase subunit 1 (Cox1) and ATP synthase F1, delta subunit (ATP5D) was up-regulated in *C. hexasticha* compared with *C. uncinata*. The oxidative phosphorylation process was also enriched in *C. hexasticha*. The main mitochondrial metabolic pathways in *C. hexasticha* were depicted and enzymes related to energy metabolism pathways were compared between these two species. More importantly, mitochondrial division inhibitor 1 (mdivi-1) proved to be very effective in killing both *C. hexasticha* and *C. uncinata*, which could be a novel drug for Chilodonellosis control. This study can help us better understand the energy metabolisms of *C. hexasticha* and *C. uncinata* and provide new insight into novel targets for chilodonellosis control. Meanwhile, the transcriptome data can also facilitate genomic studies of these two species in the future.

## 1. Introduction

Parasitic protozoa represent an important and diverse array of unicellular organisms. Among them, parasitic ciliates are considered to be among the most harmful parasites of fish, causing mass mortalities and thus substantial economic losses to aquaculture [1,2,3,4]. *Chilodonella hexasticha*, an obligate ciliate parasite, can infest a wide range of freshwater fishes and cause severe skin ulceration and gill damage, and thus high mortality [5,6]. Its congeneric species *C. uncinata* is a facultative parasite; it can be free-living or parasitize on the host fish and cause the same tissue damage as *C. hexasticha* [5,7]. The transition to parasitism is commonly associated with the moderation or loss of core metabolism [8]. A notable example is that some anaerobic parasites lost the oxidative phosphorylation function and depend on substrate-level phosphorylation to generate adenosine triphosphate (ATP) [9]. Another example is that *Ichthyophthirius multifiliis* contains far fewer lineage-specific ortholog groups in signaling pathways and gene regulation compared with its free-living relatives *Tetrahymena* and *Paramecium* [10]. As for the *Chilodonella* species, significant differences in the amino acid metabolism, lipid metabolism and tricarboxylic acid (TCA) cycle were found between the free-living *C. uncinata* and parasitic *C. uncinata* [11]. 

Mitochondria are known as the powerhouse of cells and play important roles in many cellular processes. They constantly undergo repeated fission and fusion responses to nutrient alterations, known as mitochondrial dynamics [12,13]. The crucial factor of mitochondrial fission, the GTPase dynamin-related protein 1 (Dnm1)/dynamin-related protein (Drp1) is very conserved [14]. Mitochondrial division inhibitor 1 (mdivi-1) has been proven to be a specific inhibitor of Dnm1/Drp1 and it was identified from a chemical screen for compounds that can alter mitochondrial morphology [15]. Evidence shows that mdivi-1-derived Drp1 inhibition or Drp1 knockdown can reduce cancer cell proliferation and increase spontaneous apoptosis [16]. In cells, mdivi-1 retards apoptosis by inhibiting mitochondrial outer membrane permeabilization. In vitro, mdivi-1 potently blocks Bid-activated Bax/Bak-dependent cytochrome c release from mitochondria [15]. But in pathogenic yeast, *Candida albicans*, mdivi-1 not only can repress *C. albicans* hyphal morphogenesis but also can trigger extensive metabolic reprogramming and reduce endogenous nitric oxide levels [17]. Based on the above findings, does mdivi-1 have the potential to act as an antiparasitic agent? To answer this question, the efficacy of mdivi-1 in killing *C. hexasticha* and *C. uncinata* has been evaluated in this study.

In addition, we conducted a comparative transcriptomics study of the pathogenic ciliate *C. hexasticha* and the facultative parasite *C. uncinata* based on the single-cell Smart-seq2 amplification. Non-Redundant (NR), Clusters of Orthologous Genes (COG), Swiss-Prot and Pfam databases were used for the transcriptome annotation. Gene Ontology (GO) enrichment and Kyoto Encyclopedia of Genes and Genomes (KEGG) pathway analyses were also performed to study energy-related metabolic pathways. This study can help us better understand the metabolic characteristics of *Chilodonella* species and the mechanisms underlying their adaptation to parasitism while also providing novel insights into Chilodonellosis control.

## 2. Results

### 2.1. Transcriptome Assembly Results

In total, three transcriptomes of *C. hexasticha* and *C. uncinata* were assembled in this study (Table 1). Reads obtained in this study were deposited in GenBank under the BioProject ID PRJNA910978. The clean reads numbers of transcriptomes of these two species were 5,173,514 and 18,000,967 respectively. The clean base number of *C. hexasticha* was 13, 955, 450, while the clean base numbers of free-living and parasitic *C. uncinata* were 20,617,209 and 17,746,882, respectively. Following the assembly, 36,199 and 35,534 contigs were obtained for the *C. uncinata* and 24,610 contigs were obtained for *C. hexaticha*. The GC contents of these two species were 43.66% and 48.28%, respectively. The N50 length of *C. hexaticha* was 683 bp, while the N50 lengths of free-living and parasitic *C. uncinata* were 747 bp and 717 bp, respectively. The completeness of the three transcriptomes was 60.8%, 60.8% and 73%, respectively.

### 2.2. COG Annotation

In the COG annotation results, there were 3979 unigenes identified in the *C. hexasticha*, while there were 9245 and 4964 unigenes identified in the free-living and parasitic *C. uncinate,* respectively (Figure 1A). The number of unigenes found in all three transcriptomes was 1471, and the number of unigenes that were only found in *C. hexasticha* and parasitic *C. uncinata* was 359 (Figure 1B). Among these annotated genes, the highest numbers of genes belonged to the translation, ribosomal structure and biogenesis categories, while no genes were assigned to the extracellular structures category. At the metabolism level, the number of genes assigned to the energy production and conversion in the free-living *C. uncinata* (n = 713) was twice as large as in parasitic *C. uncinata* (n = 364) and free-living *C. hexasticha* (n = 312). Compared with the free-living *C. uncinata*, *C. hexasticha* and parasitic *C. uncinata* had more similar numbers of genes assigned to other metabolic categories, such as carbohydrate transport and metabolism, amino acid transport and metabolism, lipid transport and metabolism, and coenzyme transport and metabolism. The number of genes assigned to the RNA processing and modification, and chromatin structure and dynamics categories was almost the same among the three transcriptomes. 

### 2.3. Ortholog Identification, Annotation

In total, 2151 orthologs were found between *C. hexasticha* and *C. uncinata*. Among them, 300 were up-regulated and 330 were down-regulated in *C. hexasticha* compared with the free-living *C. uncinata* (Figure 2A,D). When compared with the parasitic *C. uncinata*, 333 orthologs were up-regulated and 339 orthologs were down-regulated in *C. hexasticha* (Figure 2B,E). Meanwhile, 88 up-regulated orthologs and 108 down-regulated orthologs were identified in the parasitic *C. uncinata* compared with the free-living *C. uncinata* (Figure 2C,F).

According to the NR annotation (Figure 3), most DEGs were functionally annotated to the energy metabolism. Compared with free-living *C. uncinata* and parasitic *C. uncinata*, enzymes related to the iron-sulfur cluster (ISC) assembly, such as iron-sulfur cluster assembly scaffold protein (ISCU) and 2Fe-2S cluster-binding protein (ISCB), were up-regulated in *C. hexasticha*. Some energy metabolism-related enzymes, such as isocitrate dehydrogenase (IDH), cytochrome *c* oxidase subunit 1 (Cox 1) and ATP synthase, were also up-regulated in *C. hexasticha*. 

### 2.4. GO Enrichment Annotation 

Among the DEGs between *C. hexasticha* and free-living *C. uncinata*, 393 genes were assigned to the molecular function category, 1031 genes were assigned to the cellular component, and 2610 genes were assigned to the biological process (Figure 4A). The numbers of DEGs between the *C. hexasticha* and the parasitic *C. uncinata* assigned to the above three GO terms categories was 375, 731 and 2722, respectively (Figure 4B). In the comparison of *C. hexasticha* with both free-living and parasitic *C. uncinata*, the cellular nitrogen compound metabolic process was the most abundant subcategory within the biological process category. Other abundant subcategories included the amide metabolic process, peptide biosynthetic process, peptide metabolic process, and translation (Figure 4A,B purple part). In the molecular function category, the top three significant GO terms were binding, organic cyclic compound binding, and heterocyclic compound binding (Figure 4A,B yellow part). In the cellular component category, protein-containing complex, non-membrane-bounded organelle, and cytosol were the top three subcategories and each of them was enriched with more than 60 genes (Figure 4A,B green part). 

### 2.5. KEGG Pathway Annotation

In the comparison between *C. hexasticha* and the two *C. uncinata* types (free living and parasitic), genetic information processing, translation, and ribosome pathways were the top three KEGG pathways in which there were annotated genes. Except for the genetic information-related pathways, DEGs between *C. hexasticha* and the free-living *C. uncinata* were also annotated to the exosome and phagosome pathways (Figure 5A). DEGs between *C. hexasticha* and the parasitic *C. uncinata* were annotated to the exosome, proteasome, and oxidative phosphorylation pathways (Figure 5B). 

### 2.6. Metabolic Mitochondrial Pathways Prediction

Mitochondrial metabolic pathways of *C. hexasticha* were predicted based on the transcriptome data. These pathways included fatty acid metabolism, amino acid metabolism, pyruvate metabolism and the TCA cycle (Figure 6). 

According to the mitochondrial metabolism of *C. uncinata* reported before [18], both *C. hexasticha* and *C. uncinata* have all enzymes required for the TCA cycle. As for the electron transfer chain (ETC) complexes, six subunits (NDU5, NDUFS1, NDUFS2, NDUFS8, NDUFV1 and NDUFV2) of Complex I were identified in *C. hexasticha*, while five Complex I subunits (NDUFS1, NDUFS7, NDUFS8, NDUFV1 and NDUFV2) were identified in *C. uncinata*. The Complex II subunits SdhA and SdhB and the complex III subunits cytochrome c1 were found in both *C. hexasticha* and *C. uncinate*, and Rieske iron-sulfur protein was only found in *C. hexasticha.* Among the complex IV subunits, Cox1, Cox11, Cox15 and Cox19 were identified in *C. hexasticha*, while Cox1, Cox2, Cox6a, Cox19 and Sco1 were identified in *C. uncinata*. Among the complex V subunits, Atp1, Atp2, Atp3, Atp5 and Atp9 were detected in *C. hexasticha*, while Atp1, Atp2, Atp3, Atp5, Atp9, Atp16 and Atp20 subunits were detected in *C. uncinata*. 

### 2.7. Cell Death Assay

The mortality rate of *C. hexasticha* and *C. uncinata* in the mdivi-1 treated groups was 97% (248/255) and 95% (364/381), respectively. Significant morphological changes were observed in the mdivi-1 treated groups (Figure 7). Both *C. hexasticha* and *C. uncinata* showed shortened body length (Figure 8, Appendix A) and elongated body width and oval shape. The body length and length–width ratio of *C. hexasticha* showed significant differences between the control group and mdivi-1 treated group (Figure 8A,C), while body length, width, and length–width ratio of *C. uncinata* all showed significant differences between the two groups (Figure 8D–F).

Mitotracker staining showed that the mitochondrial size of *C. hexasticha* and *C. uncinata* became larger after mdivi-1 treatment (Figure 9 and Figure 10), which could be a sign that mdivi-1 induced mitochondrial fusion. The macronucleus had no significant differences between the control group and the mdivi-1 treated group (50 μM, 30 min) (Figure 9 and Figure 10), but nuclear lysis was also observed as cells died.

## 3. Discussion

### 3.1. TCA Cycle

The mitochondria of parasitic protists have undergone varying degrees of adaptive evolution. For instance, H_2_-producing mitochondria in *Nyctotherus ovalis* and *Blastocystis* species lack ETC complexes III and IV and only have a partial TCA cycle [19,20]. The major driving force for the transformation is considered to be oxygen, but the effect of nutrient availability cannot be neglected [21]. In this study, we found that mitochondrial oxidative phosphorylation-related pathways were enriched in *C. hexasticha*. Similarly, the TCA cycle-related enzyme, IDH, was also up-regulated in *C. hexasticha*. These findings indicate that the obligate parasitic *C. hexasticha* may have more chance to feed themselves compared with the facultative *C. uncinata*. Studies have shown that IDH in the malaria parasite *Plasmodium falciparum* can catalyze the oxidative decarboxylation of isocitrate, producing α-ketoglutarate [22,23]. IDH was also found to be functional in Leishmania parasites [24]. We hypothesize that the up-regulated IDH may play an important role in the metabolism of the obligate parasite *C. hexasticha* and it should be explored by further molecular experiments. 

All enzymes essential to the TCA cycle were identified in *C. hexasticha*, the free-living *C. uncinata*, and the parasitic *C. uncinata*. Similarly, these enzymes have also been found in the free-living ciliates *T. thermophila* and *Oxytricha trifallax* [25,26]. These TCA cycle-related enzymes are indispensable in each aerobic organism requiring oxygen for metabolism. They may be dispensable in anaerobic ciliates. For example, in anaerobic *N. ovalis*, only succinate dehydrogenase (SDH), succinyl-CoA synthetase (SCS) and malate dehydrogenase (MDH) have been identified, while MDH was proven to be cytosolic, and not involved in the mitochondrial metabolism [20]. Similar incomplete TCA cycles also exist in *Blastocystis* and *Trichomonas* [19,27]. 

### 3.2. Electron Transport Chain (ETC) Complexes

Electron transport chain (ETC) complexes are distributed in the mitochondrial inner membrane where they partake in oxidative phosphorylation. ETC uses electron transport to generate a proton gradient and couple this with the synthesis of adenosine triphosphate (ATP) [28]. The electrons are donated by the NADN and FADH_2_ generated in the TCA cycle and then transferred through Complexes I to IV [29,30]. 

In the present study, we compared all subunits of ETC Complexes I to V of *C. hexasticha* and *C. uncinata*. We found that Cox 1 was up-regulated in *C. hexasticha,* compared with *C. uncinata*. Meanwhile, the ISCU and ISCB, which are responsible for the iron-sulfur cluster biosynthesis, were also up-regulated in *C. hexasticha,* compared with *C. uncinata*. The ETC complexes contain an Iron–sulfur cluster. Thus, it may be a clue that the ETC activity in obligate parasitic *C. hexasticha* is more active, compared with facultative parasitic *C. uncinata*. We speculate that mitochondria or mitochondria-related organelles (MROs) may play an important role in parasitism. For example, MROs in the parasitic ciliate *Blastocystis* have undergone secondary rearrangements that facilitated adaptation to the low-oxygen conditions in the human gut [21,31]. Hydrogenosomes in *Trichomonas vaginalis* may contribute to parasitic preadaptation [21,32].

### 3.3. Pyruvate Metabolism

The key enzyme in pyruvate metabolism, pyruvate dehydrogenase (PDH), was found in both *C. hexasticha* and *C. uncinata*. In a typical mitochondrial metabolism, PDH catalyzes the oxidative decarboxylation of pyruvate, with the formation of acetyl-CoA, CO2 and NADH (H^+^) [33]. The acetyl-CoA is then fed into the TCA cycle. In this study, enzymes needed for the complete TCA cycle were identified in both *C. hexasticha* and *C. uncinata*. Pyruvate produced by glycolysis was transported into mitochondria by the mitochondrial pyruvate carrier (MPC), and then it was converted into acetyl-CoA by PDH. The converted products participate in the TCA cycle as substrates and thus produce energy for the cell metabolism in *Chilodonella*. PDH was also detected in *N. ovalis*, although this species does not have a complete TCA cycle [20]. In *N. ovalis*, the acetyl-CoA generated by PDH is further catalyzed by the Acetate:succinate CoA transferase (ASCT) and SCS, and thus produce energy. A PDH-encoding gene was also identified in the genome of *Blastocystis* [19]. However, in *Trichomonas*, pyruvate is decarboxylated by the pyruvate:ferredoxin oxidoreductase (PFO) [34].

### 3.4. The Fatty Acid Metabolism

In the fatty acid metabolism pathways, long-chain acyl-CoA synthetase (ASCL) and solute carrier family 25 (SLC25) were detected in both *C. hexasticha* and *C. uncinata*. ASCL activates the oxidation of fatty acids. This enzyme is present in all organisms from bacteria to humans [35]. Meanwhile, SLC25 is responsible for the transport of nutrients across the inner mitochondrial membrane for energy conversion [36]. Two enzymes that help with the fatty acid β-oxidation, 3-hydroxyacyl-CoA dehydrogenase (HAD) and enoyl-CoA hydratase (ECH) [37,38], were also detected in *C. hexasticha.* These identified enzymes may suggest that fatty acid metabolism is very important in *C. hexasticha*. Along with the numerous lipid droplets found in *C. uncinata* in our previous study [18], it can be speculated that *C. hexasticha* also stores lipid droplets as an energy supply. This can be proven by using a transmission electron microscope in the future. Unfortunately, mass culturing and sampling of *C. hexasticha* is still very challenging. 

### 3.5. Amino Acid Metabolism

Except for carbohydrates and fats, proteins also can be utilized to synthesize energy for cells. Glutamate dehydrogenase (GDH) and branched-chain amino acid aminotransferase (BCAAT) were found in both *C. hexasticha* and *C. uncinata*. GDH metabolizes α-ketoglutarate to glutamate, while BCAAT is responsible for the valine, leucine, and isoleucine metabolism. These enzymes were also found in *Blastocystis* [19]. Last but not least, enzymes related to protein transport were also detected in *C. hexasticha* and *C. uncinata*. Research has shown that HSPs help with the parasitism and differentiation of protozoan parasites [39,40,41]. In *C. hexasticha* and parasitic *C. uncinata*, HSPs may be associated with adaptation to parasitism as well.

### 3.6. Mdivi-1 for the Chilodonellids Control

In this study, 50 μM mdivi-1 showed significant efficacy in killing both *C. hexasticha* and *C. uncinata*. Mdivi-1 has been proven to be an efficacious inhibitor of mitochondrial division in yeast and can alter mitochondrial morphology [15]. It also can alter cellular oxygen consumption and induce cell death in cancer cells and inhibit complex I in mammalian cells [16,42]. In the human yeast pathogen *Candida albicans*, mdivi-1 impacts endogenous nitric oxide (NO) levels and inhibits the key virulence property of hyphal formation [17]. Mdivi-1 acts against the evolutionarily conserved mitochondrial fission GTPase Dnm1/Drp1 [14]. Drp1 homologs have been found in Opisthokonta, algae, and *Toxoplasma gondii* [43,44].

But there were no reports about the effect of mdivi-1 on parasites or ciliates. Based on the result of mdivi-1 on *C. hexasticha* and *C. uncinata* in this study, it showed that mdivi-1 could be a potential novel molecular drug for Chilodonellids control. More in-depth molecular experiments should be conducted to investigate how mdivi-1 affects *C. hexasticha* and *C. uncinata*. 

## 4. Materials and Methods

### 4.1. Specimen Collection and Identification

*Chilodonella hexasticha* were collected from the goldfish (*Carassius auratus*), purchased from an ornamental fish market in Wuhan, Hubei Province, China, in May 2018. All experimental procedures and animal care were conducted according to the National Research Council’s Guide for the Care and Use of Laboratory Animals and were approved by the Institutional Animal Care and Use Committee of the Institute of Hydrobiology, Chinese Academy of Sciences (IHB/LL/2020025). Goldfish were anesthetized with 0.02% tricaine methane sulfonate (MS-222, Sigma, St. Louis, MI, USA) according to the manufacturer’s instructions and the Chinese animal welfare regulations. Gills and fins were isolated into Petri dishes and examined with a stereoscopic microscope Stemi SV6/AxioCam MRc5 (Zeiss, Oberkochen, Germany). *Chilodonella* specimens were collected with Pasteur micropipettes, washed three times using sterilized distilled water to remove potential contaminations, flash frozen in liquid nitrogen, and then stored at −80 °C. More than five cells were collected in each sample. For morphological identification, silver nitrate staining was performed on dry smears [45]. 

### 4.2. Transcriptome Amplification, Sequencing and Assembly

The single-cell samples of *C. hexasticha* were amplified by the Smart-Seq2 method. First-strand cDNA synthesis was performed using an Oligo-dT primer, followed by PCR amplification to enrich the cDNA, and Magbeads purification to clean up the production. Then the cDNA yield was checked by Qubit^®^ 3.0 Fluorometer and Agilent 2100 Bioanalyzer (Agilent, Santa Clara, CA, USA) to ensure the expected yield with a length of 1~2 kbp. After that, the cDNA was sheared randomly by ultrasonic waves for the Illumina library preparation protocol including DNA fragmentation, end-repair, 3′ ends A-tailing, adapter ligation, PCR amplification, and library validation. After the library preparation, PerkinElmer LabChip^®^ GX Touch (PerkinElmer, Boston, MA, USA) and Step OnePlus™ (Applied BioSystems, Foster City, CA, USA) Real-Time PCR System experiments were performed for library quality inspection. Qualified libraries were then loaded in the Illumina Hiseq platform (Illumina, San Diego, CA, USA) for PE150 sequencing.

The single-cell transcriptional reads of *C. uncinata* were sequenced by us before, so we downloaded it from the NCBI website via the BioProject number PRJNA842862. In total, three samples of two *Chilodonella* species were collected and sequenced. Raw reads were trimmed with Trimmomatic v0.39 [46], processed with FastQC v0.11.9 [47], and then assembled into transcripts using Trinity v2.13.2 [48]. Transcripts were decontaminated by BLAST [49] and accessed using BUSCO v5.2.2 [50] with the alveolate_odb 10 data set.

### 4.3. Clusters of Orthologous Groups (COG) Annotation

COG annotation was conducted based on the transcripts assembled above. First COG database was downloaded from the NCBI website, then we used the makeblastdb command to build a database for the BLAST program. Next, BLASTx [49] analysis was conducted with the e-value 1e-5 to find the best hits for the transcripts. Finally, annotated proteins were analyzed and visualized using the ggplot2 package in R [51].

### 4.4. Ortholog Identification and Annotation of Differentially Expressed Genes (DEGs) 

One-to-one orthologs between *C. hexasticha* and *C. uncinata* were identified using the reciprocal best-hit method in BLASTn [49], with an e-value cutoff of 1 × 10^−5^ and a minimum identity of 50%. Only the longest transcript was chosen if there was more than one best-hit result. Orthologous protein sequences were then used for the annotation of differentially expressed genes (DEGs).

For the DEGs annotation, reads were first aligned using Bowtie v1.3.1 [52], then the expression of each transcript was quantified using RSEM [53], and a gene expression matrix was constructed. Then DEGs were identified using edgeR v3.36.0 [54], for which calcNormFactors was used to calculate normalization factors, estimateDisp was used to estimate common dispersion, and exactTest was used to determine differential expression. The read count for each predicted gene was normalized as counts per million (CPM). The adjusted *p*-value, i.e., q-value, was used for subsequent analyses. Q-value < 0.05 and |log2 fold change (FC)| ≥ 1 were set as the thresholds for the identification of differentially expressed genes. Identified DEGs were then annotated using GO terms and the KEGG pathway enrichment analyses, with the help of the online tool EggNOG-mapper (http://eggnog-mapper.embl.de (accessed on 16 October 2022)) [55]. These DEGs were further annotated using the NCBI’s nonredundant (NR) and Swiss-Prot databases. All results were visualized in R v4.1.1 [51].

### 4.5. Comparison of the Mitochondrial Metabolism

To better understand the metabolic differences between *C. hexasticha* and *C. uncinata*, mitochondria-related metabolic pathways were predicted. First, mitochondrial proteins were predicted based on the BLASTP procedure [49] and the well-described mitochondrial proteomes of *Homo sapiens* [56], *Saccharomyces cerevisiae* [57] and *Tetrahymena thermophila* [26] were used as queries. Second, blast results were further annotated by the BlastKOALA (www.kegg.jp/blastkoala/ (accessed on 16 October 2022)) server [58]. Then, TargetP [59] and MitoFates [60] were used to predict the mitochondrial-targeting signals. Finally, metabolic pathways were visualized using Adobe Illustrator.

### 4.6. Mdivi-1 Evaluation Assay

To find potential novel targets for chilodonellosis control, mdivi-1 was used in this study and its metabolic effect was examined. *C. hexasticha* and *C. uncinata* cells were collected from the gills of goldfish and culture medium, respectively. Cells were placed into dishes and treated with DMSO, mdivi-1 (50 μM) for 30 min and then collected and stained by 0.04% trypan blue and incubated for 3 min. Each treatment was repeated three times. Then the dead cells (stained) and live cells (unstained) were counted. And the mortality rate was calculated to evaluate the efficacy of mdivi-1 against *C. hexasticha* and *C. uncinata*. The body length, width and length–width ratio were measured (n > 25 in each group) and compared using a *t* test. 

To explore the effects of the mdivi-1 on mitochondria, cells were also incubated with Mitotracker Red CMXRos (200 nM; Meilun Biotechnology, Dalian, China) for 25 min at room temperature. DAPI (1 μg/mL; Genview, Beijing, China) was also used to locate the nucleus. Pictures were taken with a microscope (ZEISS Axio Imager A2, Carl Zeiss, Jena, Germany) equipped with a digital camera (sCMOS PCO, Kelheim, Germany).

## 5. Conclusions

In summary, the energy metabolic pathways of *C. hexasticha* were analyzed in this study. The main mitochondrial metabolic pathways, such as the TCA cycle, ETC and pyruvate metabolism, etc. were compared between *C. hexasticha* and *C. uncinata*. Additionally, the effectiveness of mdivi-1 as a potential targeted drug was evaluated. This study increased the amount of molecular data available for these two parasitic ciliate species and helped us better understand the energy metabolism and parasitism adaptation in Chilodonellids. This study also provided new ideas for Chilodonellosis control.

## Figures and Tables

**Figure 1 ijms-24-13058-f001:**
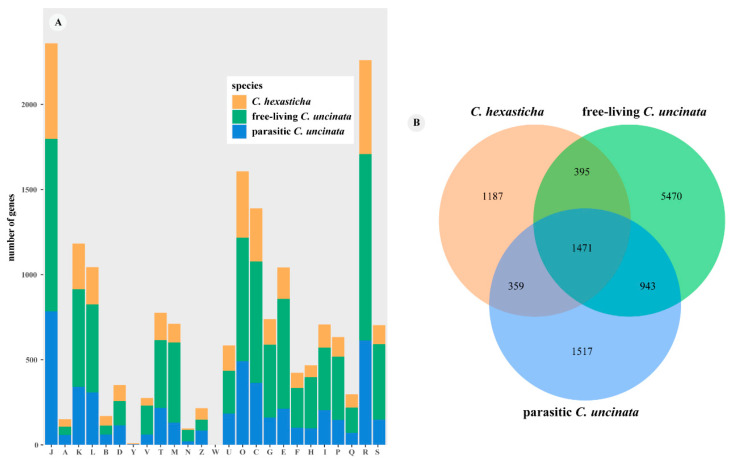
COG annotation of the three transcripts of *C. hexasticha* and *C. uncinata*. For specific data, see Appendix A. (**A**). COG annotation results. J—Translation, ribosomal structure and biogenesis, A—RNA processing and modification, K—Transcription, L—Replication, recombination and repair, B—Chromatin structure and dynamics, D—Cell cycle control, cell division, chromosome partitioning, Y—Nuclear structure, V—Defense mechanisms, T—Signal transduction mechanisms, M—Cell wall/membrane/envelope biogenesis, N—Cell motility, Z—Cytoskeleton, W—Extracellular structures, U—Intracellular trafficking, secretion, and vesicular transport, O—Posttranslational modification, protein turnover, chaperones, C—Energy production and conversion, G—Carbohydrate transport and metabolism, E—Amino acid transport and metabolism, F—Nucleotide transport and metabolism, H—Coenzyme transport and metabolism, I—Lipid transport and metabolism, P—Inorganic ion transport and metabolism, Q—Secondary metabolites biosynthesis, transport and catabolism, R—General function prediction only, S—Function unknown; (**B**). Venn diagram of COGs between *C. hexasticha* and two *C. uncinata*.

**Figure 2 ijms-24-13058-f002:**
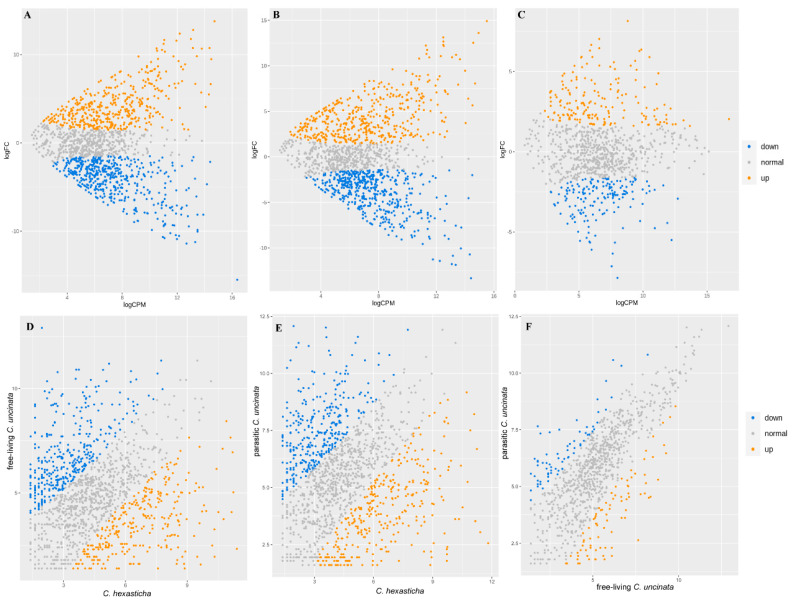
MA-plot and scatter-plot of *C. hexasticha* and *C. uncinata* showing differentially expressed genes based on one-to-one ortholog results. (**A**). DEGs between *C. hexasticha* and the free-living *C. uncinata*; (**B**). DEGs between *C. hexasticha* and the parasitic *C. uncinata*; (**C**). DEGs between the free-living *C. uncinata* and the parasitic *C. uncinata.* (**D**). The count of DEGs counts between *C. hexasticha* and the free-living *C. uncinata*; (**E**). The count of DEGs counts between *C. hexasticha* and the parasitic *C. uncinata*; (**F**). The count of DEGs counts between the free-living *C. uncinata* and the parasitic *C. uncinata*.

**Figure 3 ijms-24-13058-f003:**
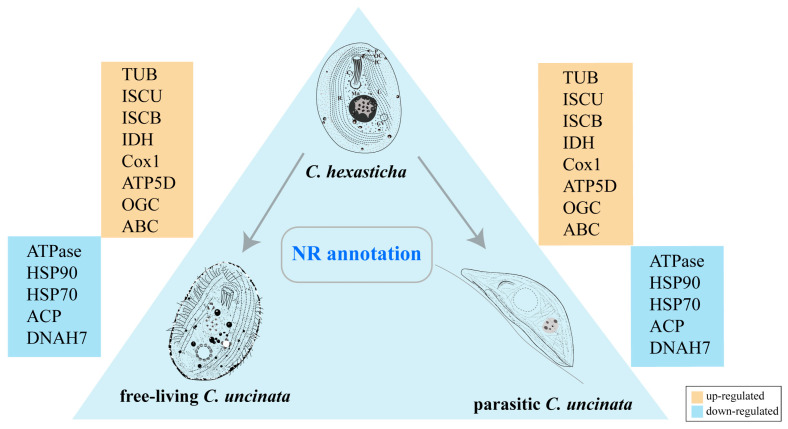
NR annotation results of DEGs between *C. hexasticha* and *C. uncinata.* The arrows indicate a comparison between two species. The schematic representation of *C. hexasticha* was hand drawn for this study, while those of *C. uncinata* were adopted from [7]. The yellow boxes represent the up-regulated genes, while the blue boxes represent down-expressed genes. The full names of protein products of these annotated genes are TUB: tubulin, ISCU: iron-sulfur cluster assembly scaffold protein IscU-like; ISCB: 2 iron, 2 sulfur cluster-binding protein, IDH: isocitrate dehydrogenase, Cox1: cytochrome c oxidase subunit 1, ATP5D: ATP synthase F1, delta subunit, OGC: 2-oxoglutarate/malate carrier protein, ABC: ABC transporter family protein, ATPase: v-type ATPase subunit family protein, HSP: heat shock protein, ACP: acyl carrier protein, DNAH7: dynein heavy chain 7.

**Figure 4 ijms-24-13058-f004:**
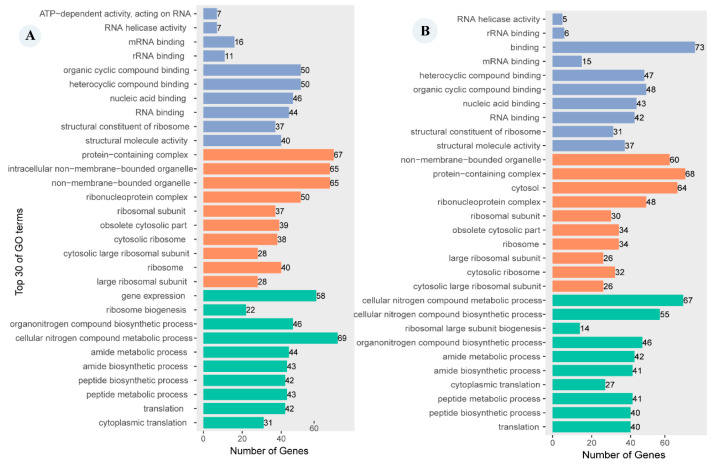
Top 30 GO terms enrichment results of DEGs between *C. hexasticha* and *C. uncinata.* For specific data, see Appendix A. Purple represents the molecular function category, yellow represents the cellular component category, and green represents the biological process category. (**A**). DEGs between *C. hexasticha* and free-living *C. uncinata*. (**B**). DEGs between *C. hexasticha* and parasitic *C. uncinata*.

**Figure 5 ijms-24-13058-f005:**
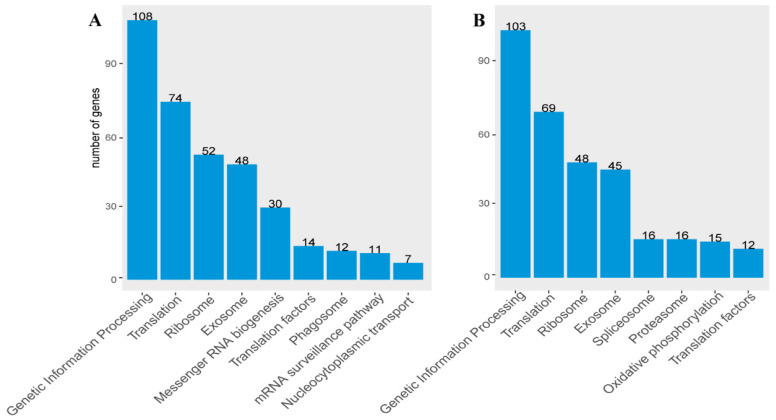
KEGG pathway enrichment of DEGs between *C. hexasticha* and *C. uncinata.* For specific data, see Appendix A. (**A**). DEGs between *C. hexasticha* and the free-living *C. uncinata*; (**B**). DEGs between *C. hexasticha* and the parasitic *C. uncinata*.

**Figure 6 ijms-24-13058-f006:**
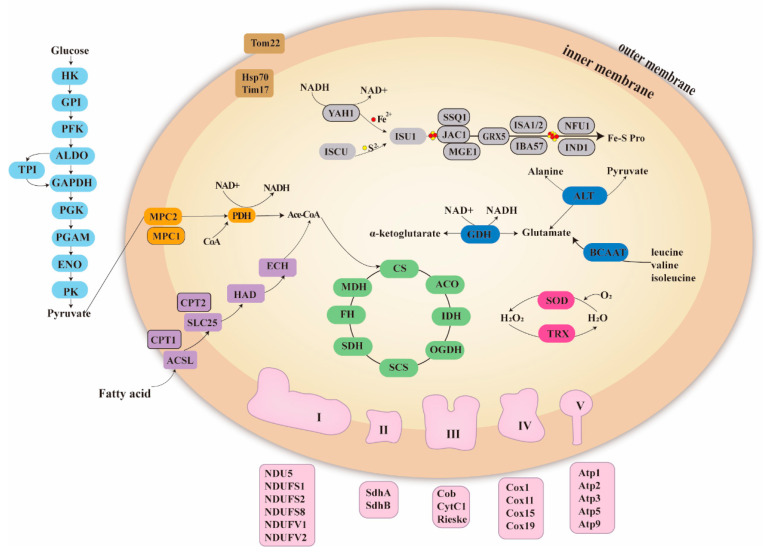
Mitochondrial metabolism in *C. hexasticha.* For full names and other information about these abbreviated proteins see Appendix A. Colors are used to indicate metabolic pathways. Light blue: glycolysis, orange: pyruvate metabolism, purple: fatty acid metabolism, green: TCA cycle, pink: ETC complexes, red: superoxidation reaction, dark blue: amino acid metabolism, grey: Fe-S cluster assembly, brown: protein import system.

**Figure 7 ijms-24-13058-f007:**
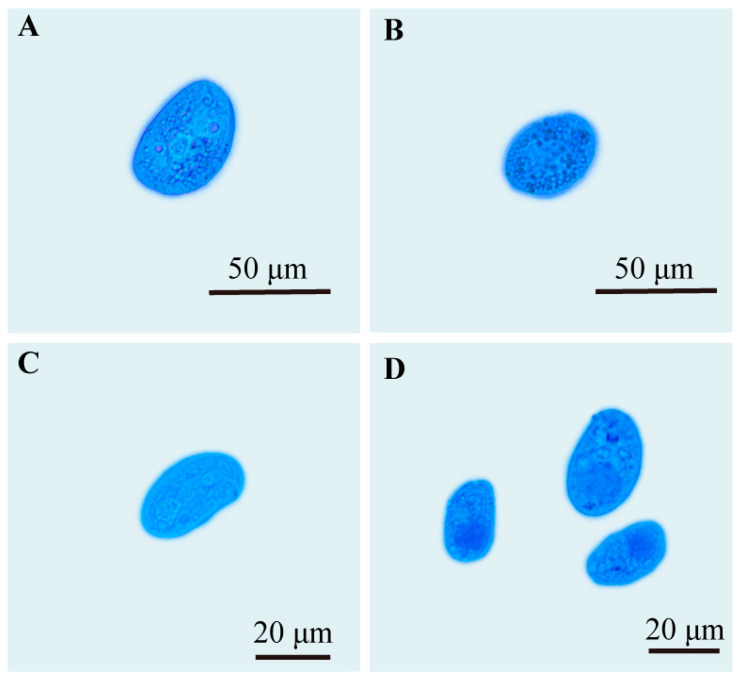
Trypan blue staining of *C. hexasticha* and *C. uncinata*. (**A**,**B**): *C. hexasticha* treated with DMSO and 50 μM mdivi-1 for 30 min, respectively; (**C**,**D**): *C. uncinata* treated with DMSO and 50 μM mdivi-1 for 30 min, respectively.

**Figure 8 ijms-24-13058-f008:**
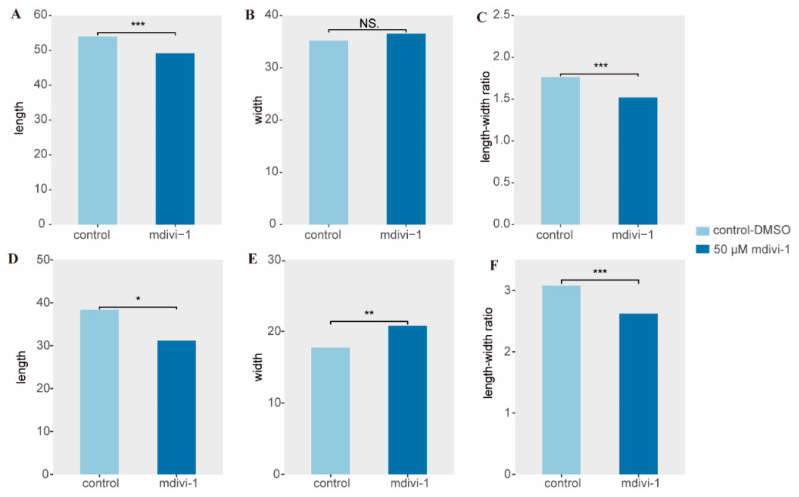
Measurement data of *C. hexasticha* and *C. uncinata* in control group and mdivi-1 treated group. (**A**–**C**): body length, body width, and length–width ratio of *C. hexasticha,* respectively; (**D**–**F**): body length, body width, and length–width ratio of *C. uncinata,* respectively. * means *p*-value < 0.05, ** means *p*-value < 0.01, *** means *p*-value < 0.001, and NS. means *p*-value of no significant difference.

**Figure 9 ijms-24-13058-f009:**
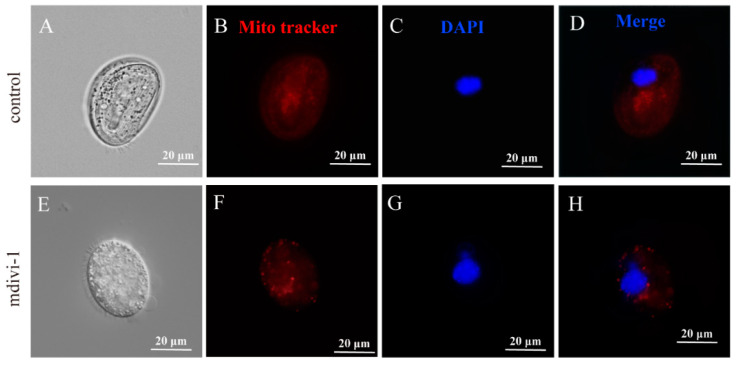
Mdivi-1 altered the size of mitochondria in *C. hexasticha*. (**A**,**E**). Light microscope images. (**B**,**F**). Mito tracker staining to show the mitochondria in red. (**C**,**G**). DAPI staining to show the nucleus in blue. (**D**,**H**). merge of fig B and C, and fig F and G, respectively.

**Figure 10 ijms-24-13058-f010:**
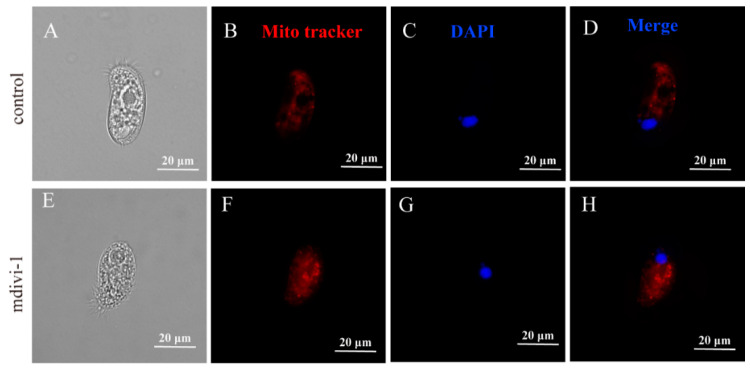
Mdivi-1 altered the size of mitochondria in *C. uncinata*. (**A**,**E**). Light microscope images. (**B**,**F**). Mito tracker staining to show the mitochondria in red. (**C**,**G**). DAPI staining to show the nucleus in blue. (**D**,**H**). merge of fig B and C, and fig F and G, respectively.

**Table 1 ijms-24-13058-t001:** Basic information about sequencing and assembly.

Species	Clean Reads Number	Clean Bases	GC	N50	Mapping Rate	Contigs Number	Average Length	BUSCO
*C. hexasticha*	5,173,514	13,955,450	43.66	683	98.83%	24,610	620.63	60.8%
free-living *C. uncinata*	7,595,875	20,617,209	47.46	747	98.29%	36,199	650.57	60.8%
parasitic *C. uncinata*	10,405,092	17,746,882	49.09	717	98.81%	35,534	637.80	73%

## Data Availability

The raw paired-end reads in this study were deposited in the NCBI SRA with BioProject ID PRJNA910978.

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
