# Peer review of "Comparative Transcriptomics of Chilodonella hexasticha and C. uncinata Provide New Insights into Adaptations to a Parasitic Lifestyle and Mdivi-1 as a Potential Agent for Chilodonellosis Control"

_ijms, 2023, doi:10.3390/ijms241713058_

Round 1
Reviewer 1 Report
Abstract.
mdivi-1: please report the full name.
Introduction:
mdivi-1: it lacks a deeper description of this molecule and its activity
line 50: rephrase the statement, it is not good to start with "and".
REsult
Fig. 2 is not readable and difficult to understand.
Fig. 3 also is not clear: are exactly the same genes up and down regulated?
The paper lacks a test of the mdivi-1 on non parasitic ciliate. From the paper it appears that mdivi-1 acts exactly on both ciliate species and it is not specific for the obligatory parasite.
It can be improved.
Reviewer 2 Report
Dear Editor,
The manuscript entitled “Comparative transcriptomics of Chilodonella hexasticha and C. uncinata provide new insights into adaptations to a parasitic lifestyle and mdivi-1 as a potential agent for chilodonellosis control” by Xia-lian Bu et al. presents a single-cell (parasite) transcriptomic study of Chilodonella hexasticha and uncinata, focusing on enzymes related to energy metabolism and parasitic adaption.
Τhe manuscripts’ objects are interesting, it is well written in a comprehensive way and the findings are interesting. Therefore, the manuscript could be accepted for publication after major revisions:
1. In my opinion the major drawback of the manuscript is that the authors analyze only one biological sample per situation as seen in section 2.1, i.e. one sample for C. hexasticha, one sample for free-living C. uncinata and one parasitic C. uncinata. Even though methodology and result analysis are correct, the authors should have used at least 3 biological replicated for each situation to have a scientifically accepted study.
2. Also, the discussion section needs to be enriched. In my opinion it contains much information however I suggest to add some more quantitative aspects of the findings based on the results. Maybe building a protein interaction network would be useful.
Minor suggestions:
3. Figure 2 needs to be larger to be readable.
4. The blue background should be removed from Figure 7, it is confusing.
5. In page 11, line 331: Wellborn, 1967 has not be added to the reference list.
6. Please correct the references numbering based on their appearance in the text.
Round 2
Reviewer 2 Report
Dear editor, the authors have incorporated most of my comments on their revised manuscript and I thank them. I am troubled by the fact that the authors state that “More than five cells were collected in each sample” but in section 4.2 they analyze one single cell sample per condition (as seen in table 1). Maybe I did not understand it correctly, but based on what is written the authors either pooled the 5 cells from each same and analyzed it as one or analyzed one of those cells. In both cases, the biological replicates are missing. So please clarify. Also, for my second comment I fully understand that the authors were not able to build the PPI network and I accept it. However, the comment main focus was the need to enrich the discussion with quantitative aspects of the finding which is not included in the revised manuscript
However, if the editor decide that these two points can be overlooked, I suggest publication in its present form
